# On the Implementation of a Post-Pandemic Deep Learning Algorithm Based on a Hybrid CT-Scan/X-ray Images Classification Applied to Pneumonia Categories

**DOI:** 10.3390/healthcare11050662

**Published:** 2023-02-24

**Authors:** Abdelghani Moussaid, Nabila Zrira, Ibtissam Benmiloud, Zineb Farahat, Youssef Karmoun, Yasmine Benzidia, Soumaya Mouline, Bahia El Abdi, Jamal Eddine Bourkadi, Nabil Ngote

**Affiliations:** 1MECAtronique Team, CPS2E Laboratory, National Superior School of Mines Rabat, Rabat 53000, Morocco; 2ISITS-Maintenance Biomédicale-/Rabat, Abulcasis International University of Health Sciences, Rabat 10000, Morocco; 3ADOS Team, LISTD Laboratory, National Superior School of Mines Rabat, Rabat 53000, Morocco; 4SSDT Team, LISTD Laboratory, National Superior School of Mines Rabat, Rabat 53000, Morocco; 5Medical Simulation Center/Rabat of the Cheikh Zaid Foundation, Rabat 10000, Morocco; 6Cheikh Zaïd International University Hospital, B.P. 6533, Rabat 10000, Morocco; 7Faculty of Medicine and Pharmacy, Mohammed V University, B.P. 6203, Rabat 10000, Morocco

**Keywords:** pneumonia, artificial intelligence, deep learning, EfficientNetB7, lungs, X-ray, CT scan, thoracic imaging

## Abstract

The identification and characterization of lung diseases is one of the most interesting research topics in recent years. They require accurate and rapid diagnosis. Although lung imaging techniques have many advantages for disease diagnosis, the interpretation of medial lung images has always been a major problem for physicians and radiologists due to diagnostic errors. This has encouraged the use of modern artificial intelligence techniques such as deep learning. In this paper, a deep learning architecture based on EfficientNetB7, known as the most advanced architecture among convolutional networks, has been constructed for classification of medical X-ray and CT images of lungs into three classes namely: common pneumonia, coronavirus pneumonia and normal cases. In terms of accuracy, the proposed model is compared with recent pneumonia detection techniques. The results provided robust and consistent features to this system for pneumonia detection with predictive accuracy according to the three classes mentioned above for both imaging modalities: radiography at 99.81% and CT at 99.88%. This work implements an accurate computer-aided system for the analysis of radiographic and CT medical images. The results of the classification are promising and will certainly improve the diagnosis and decision making of lung diseases that keep appearing over time.

## 1. Introduction

The world was gripped by the COVID-19 pandemic over the first half of 2020, it’s spread had severe and damaging sociological impacts and led to a significant slowdown in economic activities. In this context, it is mentioned that the respiratory virus SARS-CoV-2 (Severe Acute Respiratory Syndrome Coronavirus-2) became a major pandemic because of its easy spread in the air and contact with contaminated objects and people. In less than a year, it has plunged humanity into an unprecedented crisis that has spared no area of life [1]. So far, this pandemic has affected more than 200 countries, with more than 591 million coronavirus pneumonia cases and more than 6.4 million deaths directly linked to this coronavirus, officially counted on 19 August 2022 [2,3]. This enormous number of cases has led the medical community to take a greater interest in this kind of disease. Over time, the emergence of respiratory viruses has placed humanity in a context of uncertainty and fear, often accentuated by the urgency of the response. They affect all people of both sexes and all age groups, including smokers and non-smokers, making them one of the most widespread health problems worldwide [4]. Among the most common lung diseases are asthma, bronchitis, emphysema, lung cancer, common pneumonia, and coronavirus pneumonia. It should be noted that pneumonia caused by coronaviruses may be misdiagnosed as common pneumonia. It is an infection that affects one or both lungs. It can be caused by bacteria, viruses, or fungi. Symptoms can range from mild to severe and may include coughing either with or without mucus (a slimy substance), fever, chills, and breathing difficulty [5]. Since pneumonia caused by respiratory viruses causes the same symptoms, additional high-precision methods are needed to confirm this pneumonia’s etiology. One of the most reliable tests is the Polymerase Chain Reaction (PCR) which is recommended by the WHO [6,7]. Among the instrumental methods used, as an alternative or confirmation approach and recommended by the WHO to diagnose lung lesions, imaging can be done either by radiography, Computed Tomography (CT) or ultrasound [8]. Coronavirus pneumonia infection is not so rare in everyday life. Over time, several coronavirus cases have manifested as common pneumonia, since they both have very similar symptoms. Therefore, the main objective is to relieve the medical community and facilitate the differentiation between these diseases. Given the increasing number of cases, the use of such methods shows limitations. A computer science resource will be relevant. Besides, since the 1990s, the Artificial Intelligence (AI) industry has made significant progress. It was first described by John McCarthy in 1956 as “The science and engineering of making intelligent machines.” [9]. Nevertheless, early models’ flaws hindered widespread adoption and medical application. With the rise of Machine Learning (ML) in the 1980s and Deep Learning (DL) in 2010, many of these limitations were overcome. Thus, the ability of AI to exploit significant associations in a dataset has started to be used for diagnosis, therapy, and outcome prediction in many clinical contexts [10]. In 2013, Maoling Zhu et al. used the computer-aided diagnosis to differentiate Pancreatic Cancer (PC) from normal tissue [11]. In 2017, Gargeya et al. used DL for diabetic retinopathy screening, their model achieved 94% sensitivity and 98% specificity [12]. By the same year, AI was also applied to reliably predict cardiovascular risk, failing to recognize a large number of individuals who would benefit from preventative care while others underwent unneeded intervention. Machine Learning was used to improve accuracy by taking advantage of the complex interactions between risk factors [13]. Pneumology was no exception to this trend since many AI techniques were used both to classify and detect lung diseases. In this context, the emergence of pneumonia coronavirus around the world has challenged researchers to provide rapid and effective diagnostic tools to make healthcare systems intelligent in the fight against pandemics caused by respiratory viruses [14]. Thus, the use of AI remains mandatory, and the development of convolutional networks and DL algorithms that researchers, specialists, and companies around the world are deploying, has enabled a revolution in the rapid processing of hundreds of X-ray and CT scan images. Several algorithms to enhance, accelerate, and make an accurate diagnosis of different cases of pneumonia and aid in decision-making were developed [15]. For that, this work will present a Post-Pandemic Classification Algorithm (PPCA) in three classes: common pneumonia, coronavirus pneumonia, and healthy lungs; on two different modalities, chest X-rays and chest CT scans. This will be done using artificial intelligence tools such as the Convolution Neural Network (CNN) called EfficientNet and preprocessing techniques. The major contributions of this paper are:Using preprocessing techniques to enhance the quality of images;Training DL model to distinguish between common pneumonia, coronavirus pneumonia and normal cases;Performing classification on two different modalities including chest X-ray and chest CT scan.

Many studies have been devoted to deep learning-based solutions for detecting lung diseases. Among the studies already mentioned in the related word section, we cite: (i) Nishio et al. [16] who used pre-trained models, (VGG16, Resnet-50, MobileNet, DenseNet-121 and EfficientNet) for X-ray image classification between COVID-19 pneumonia, non-COVID-19 pneumonia and healthy individuals with an accuracy of 83.6% and an average sensitivity of COVID-19 pneumonia of 90.9%; (ii) Ucar and Korkmaz [17] proposed a Bayes-SqueezeNet based rapid diagnostic system; an accuracy performance was 98.3% (among normal, pneumonia and Covid cases), and 100% for unique COVID-19 recognition; (iii) Maftouni et al [18] presented a robust COVID-19 classifier on chest CT images by proposing a deep learning model of pre-trained Residual Attention-92 and DenseNet architectures ensemble. The results were 97.93% and 98.32% accuracy respectively with COVID-19 pneumonia sensitivity of 96.72% and 98.06%. Regarding the present study, comparing to the above mentioned studies the proposed model based on EfficientNet B7 architecture showed its effectiveness in providing better results for hybrid CT Scan and X-ray classification between COVID-19 pneumonia, non-COVID-19 pneumonia and healthy lungs, with an accuracy successively of 99.81% and 99.88% and a sensitivity to COVID-19 pneumonia of 100% for both imaging modalities.

The following is the structure of the research paper. Section 2 summarizes previous work in the field of pneumonia classification. Section 3 covers all stages of the proposed approach, from image preprocessing to image classification. Section 4 discusses the research’s implementation specifics and the experimental findings. Section 5 depicts a discussion, and finally, Section 6 closes the paper with a conclusion.

## 2. Related Work

Deep Learning (DL) technology is currently being implemented in various subfields of medicine, including diagnostics, bioinformatics, and education. Since the onset of the pandemic, many researchers have shown the effectiveness of using the concept of transfer learning in DL frameworks, even is still limited. Part of the challenge deals with distinguishing between common pneumonia, coronavirus pneumonia, and normal cases. Several DL models can be implemented in medical image classification and analysis to support speed and correct decision-making.

### 2.1. X-ray-Based Approaches

Tripathi et al. [19] proposed and evaluated a deep Convolutional Neural Network (CNN) designed to classify thoracic diseases. The proposed model consists of convolutional layers, ReLU activations, a pooling layer, and a fully connected layer. An open-source dataset called Chest X-ray 14 was used. It consists of fifteen categories called atelectasis, cardiomegaly, effusion, infiltration, mass, nodule, pneumonia, pneumothorax, consolidation, edema, emphysema, fibrosis, pleural thickening, and hernia. This model gives an average accuracy of 89.77% obtained for the classification of different diseases. The comparative analysis shows the effectiveness of the proposed model. Zhang et al. [20] developed a deep model to identify coronavirus pneumonia infection from radiological images. This model was trained on a dataset comprising radiological images from 1008 patients with common pneumonia and 70 patients with coronavirus pneumonia. It achieved a sensitivity of 96.0% and a specificity of 70.7% with an AUC of 95.2%. Sarki et al. [21] have developed a CNN-based system trained from scratch for persuasive classification and reliable detection of coronavirus pneumonia using a public X-ray database for training and validation. The data collected consisted of 1341 healthy images, 296 images with positive and suspect coronavirus pneumonia, and 3875 images with positive viral and bacterial pneumonia. Therefore, an imbalance in the collected data can be observed, which can lead to misleading classification results. Therefore, they examined all images manually and removed overexposed and underexposed images. Finally, they selected 140 images from each category for their experiments. Tuncer et al. [22] used the novel fuzzy tree classification approach for X-ray images in three classes (normal cases, common pneumonia, and coronavirus pneumonia). They applied Multi-Kernel Local Binary Pattern (MKLBP) to generate features, which were selected using the interactive neighborhood component (INCA) feature selector. INCA selected 616 features, which were forwarded to 16 conventional classifiers in five groups: Decision Tree (DT), Linear Discriminant (LD), Support Vector Machine (SVM), Ensemble Learning (EL), and K-Nearest Neighbor (K-NN). The best classifier was the cubic SVM which achieved 97.01% classification accuracy. The application was applied using the MATLAB (2019b) software, with the MATLAB Classification Learner Toolbox (MCLT) for classification. Nishio et al. [16] aimed at developing and validating a computer-aided diagnostic system for the classification of a total of 1248 chest X-ray images, including 215 with coronavirus pneumonia, 533 with common pneumonia, and 500 healthy. They used 4 pre-trained models for transfer learning. VGG16 was the most accurate for category classification with a ratio of 83.6%. As the study dataset was relatively small (1248), it was necessary to improve the robustness of the CNN models by building an accurate CNN model using both transfer learning with VGG16 and a combination of data augmentation methods. Asif et al. [5] aimed to automatically detect patients with coronavirus pneumonia using chest X-ray images while maximizing detection accuracy using deep convolutional neural networks (DCNN) on 864 cases of coronavirus pneumonia, 1345 cases of viral pneumonia, and 1341 cases of normal pneumonia. The authors used Inception-v3 with transfer learning. The classification accuracy reached more than 98%, with a training accuracy of 97% and a validation accuracy of 93%. Here, DCNN performed better with a larger dataset than with a smaller one. Shelke et al. [23] tended to apply a specific method. From an X-ray screening machine, the chest X-ray image was run through the VGG16 model, with the results bracketed as normal, pneumonia, and TB. The pneumonia images were then run through the DenseNet-161 model and classified as normal pneumonia and coronavirus pneumonia. These coronavirus pneumonia images were run through a ResNet-18 model and classified as severe, moderate, and mild Coronavirus pneumonia. The VGG16 achieved an accuracy rate of 96%, denseNet-161 reached 98.9%, and 76% for ResNet-18. Ucar and Korkmaz [17] handled the problem of imbalanced data of the public dataset by using a multiscale offline augmentation technique. After that, the authors trained the augmented data with SqueezeNet architecture. The approach achieved an accuracy of 98.3%. Elaraby et al. [24] designed a new Gray-Scale Spatial Exploitation Net (GSEN) to classify patients with coronavirus pneumonia. This approach used web page crawling across cloud computing environments. The approach achieved an accuracy of 92.76% for three-class labels and 95.60% for two-class labels. Table 1 shows a summary of the comparison between the different studies that discussed X-ray-based approaches.

### 2.2. CT-Scan Based Approaches

Saba et al. [25] used six models, namely K-NN and RF based on traditional machine learning, VGG19, and Inception-v3 based on transfer learning. Then, they used CNN and iCNN based on personalized deep learning to address the classification between coronavirus pneumonia and common pneumonia from CT images. The dataset used was 2758 coronavirus pneumonia CT scans and 990 non-coronavirus pneumonia CT scans. The customized deep learning models of CNN and iCNN gave good results, with a ratio of 99.53% for CNN and 99.69% for iCNN. Shi et al. [26] used CT scan images of 1658 patients with coronavirus pneumonia and 1027 patients with common pneumonia. The authors proposed the infection-size-aware random forest (iSARF) method. Subjects were automatically classified with different ranges of infected lesion sizes, followed by random forests within each group for classification. The results were: acc: 87%, AUC: 94%, sensitivity: 90% and specificity: 83%. The proposed method has been integrated into the Ucloud platform as an online service and is available to more than 20 clinical facilities in China. Maftouni et al. [18] presented a coronavirus pneumonia classifier on chest CT images with noisy labels by proposing an ensemble deep-learning model of pre-trained residual attention and DenseNet architectures. The novelty of this method is that the features extracted from the two deep networks (core learners) are stacked together and processed by a meta-learner to provide the final, robust prediction. The results in terms of precision were respectively 97.93% and 98.32% for the two proposed models. Nguyen et al. [27] conducted an evaluation of DL classification models trained to identify patients with positive coronavirus pneumonia on 3D computed tomography (CT) datasets from different countries: CC-CCII Dataset (China), COVID-CTset (Iran) and MosMedData (Russia) in addition to a dataset at UT Southwestern (UTSW). Models trained on a single dataset achieved receiver operating accuracy (AUC) values of 0.87/0.826 (UTSW), 0.97/0.988 (CC-CCCI) and 0.86/0.873 (COVID-CTset). Apart from this, models trained on multiple datasets and evaluated on a test set from one of the datasets used for training performed better. However, performance dropped by almost an AUC of 0.5 (random estimate) for all models when evaluated on a different dataset outside of its training datasets, including the MosMedData. Pathan et al. [28] proposed two models for Coronavirus pneumonia detection: (i) based on a transfer learning approach and (ii) using a novel strategy to optimize CNN hyperparameters using the BAT algorithm based on Whale optimization + AdaBoost classifier built using dynamic ensemble selection techniques. The proposed system achieved 96% classification accuracy in detecting Coronavirus pneumonia using chest CT scans. Table 2 shows a summary of the comparison between the different studies that discussed CT-Scan-based approaches.

Through our approach, we aim to improve the results obtained by other authors. The main objective is to create a lung disease classification model with higher accuracy that will be trained to recognize both X-ray images and CT scans.

## 3. Materials and Methods

To carry out this work, image processing was done to improve the quality of the used images. Followed by the choice and enhancement of the classification network to automatically classify pneumonia.

### 3.1. Image Preprocessing

The preprocessing pipeline is habitually used for preparing the input layer to satisfy the CNN requirements. In this work, image preprocessing was performed in four steps (Figure 1). The first step concerns only the CT scan dataset. The second step represents the image quality improvement, which is separated into two different methods: Contrast Limited Adaptive Histogram Equalization for X-ray modality and standard Histogram Equalization for CT scan modality. The third and the last steps are common for both modalities.

#### 3.1.1. Cropping the Region of Interest (RoI)

CT scan images contain a high number of unwanted and insignificant pixels. For this reason, the RoI is cropped using some image processing techniques. First, Otsu’s Method is applied to perform unsupervised image thresholding. This step provides two different thresholds that are taken into consideration for the quantization of the images. The second step is quantizing the image using two quantization levels (i.e., two values of the threshold) to obtain only two regions. Third, binarizing the image to keep only the RoI that contains the lungs as well as the surrounding thoracic tissue. As shown in Figure 2, the RoI is cropped to extract only the efficient pulmonary regions.

#### 3.1.2. Improving the Image Quality

Histogram Equalization is an image processing technique that adjusts image intensities to enhance the contrast. In this method, the probability density function of a given image is modified into a uniform probability density function which spreads out from the lowest pixel value to the highest one.

Contrast Limited Adaptive Histogram Equalization (CLAHE) aims to enhance the local contrast of an image [29]. CLAHE calculates the contrast transform function for each region individually. The contrast of each tile is enhanced so that the histogram of the output region approximately matches the histogram specified by the ’Distribution’ value. Neighboring regions are then combined using bilinear interpolation to remove artificially induced boundaries. The contrast, especially in homogeneous areas, can be limited so as not to amplify any noise present in the image. Figure 3 depicts the image quality improvement after applying the CLAHE technique, in which some lung details are more discriminant.

#### 3.1.3. Image Resizing

Deep Learning models generally train more quickly on small images. For this reason, it is necessary to resize all the images into 256×256 dimensions, to provide the most adequate dataset for the used model which will allow obtaining the best results. The method takes the input image as input and a scaling factor and scales the input image with that factor.

#### 3.1.4. Data Augmentation

To get good performance, the used model should be trained on a proportional number of examples. To both increase the number of training images and avoid problems with unbalanced data, the Augmentor tool was used. Augmentor is a Python package designed to generate artificial images for machine and deep-learning problems. For this purpose, rotational, vertical, and horizontal mirror transformations were used to generate 600 images per class for the X-ray dataset. Since the CT scan dataset is large enough to train the proposed model, the data augmentation step was skipped.

### 3.2. Classification Network Architecture

A deep learning architecture based on EfficientNetB7 which is known as the most advanced architecture in convolutional networks was built [30]. EfficientNetB7 shows a particularity in using a scaling strategy that employed a compound coefficient to equitably scale all the architecture parameters including resolution, depth, and width. First, a convolutional network is used to learn feature maps, while the second is used to classify the input images. Each convolutional layer was followed by an activation layer (ReLU- Rectified Linear Unit). By measuring a weighted sum, activation determines whether a neuron needs to be activated or not. It is used to introduce nonlinearity into the output of a neuron. In the following steps, max-pooling is performed to downsample the input image, reduce dimensionality, and prepare it for processing. The pre-trained model was used on the ImageNet dataset and the last layers of each model (1000 classes). Afterward, batch normalization, fully connected, and dropout layers were applied. As shown in Figure 4, the dropout layers are used to prevent overfitting. Initialization of the weight kernel with orthogonal weights was also performed. During the forward pass through a CNN, this initialization prevents the layers’ activation outputs from exploding or disappearing. Lastly, a fully-connected layer with three neurons was added to represent the class scores (i.e., output layer).

### 3.3. Used Datasets

#### 3.3.1. X-ray Dataset

The used dataset is composed of the publicly available datasets named MOMA Dataset and was published in June 2020 [31]. It is composed of 603 X-ray images in JPEG format, downloaded from Mendeley. 234 of these images were normal, 221 were positive for coronavirus pneumonia and 148 of them were positive for pneumonia. The pneumonia images were completed by 100 images recovered from the Cheikh Zaid International University Hospital.

#### 3.3.2. CT Scan Dataset

MosMedData [32] was acquired from 1 March 2020 to 25 April 2020 at municipal hospitals in Moscow, Russia. It consists of anonymized lung CT scans with COVID-19 signs, as well as CT scans without such findings. It contains several CT scans for 1110 patients, of whom 42% were males, 56% were females and 2% were not identified. The patient’s age ranges from 18 to 97 years old. Every exam has been saved in NIFTI (Neuroimaging Informatics Technology Initiative) format and archived in GZIP format. During this process, only every 10th instance was maintained in the final file. To complete the missing pneumonia cases for classification on 03 classes, a database named the largest COVID-19 CT dataset was sollicted [18] from [(https://www.kaggle.com/datasets/maedemaftouni/large-covid19-ct-slice-dataset) (accessed on 17 February 2022)].

#### 3.3.3. Cheikh Zaid Data

Validation images were obtained on one CT system (Somatom Def AS, Siemens Healthineers, Germany). The main scanning parameters were as follow: Tube voltage: 120 kV, pitch factor 1/4 0.3–1.5 mm, recon matrix 1/4 512×512, slice thickness 1/4 1 mm. The patients were positioned toward the front of the imaging equipment where the face is in an upward direction (i.e., Head FirstSupine). The patient’s dataset is saved in DICOM (Digital Imaging and Communication in Medicine) 3.0 format. This study is a prospective analysis approved by the ethics committee of the Cheikh Zaid International University Hospital. 127 patients suspected of pneumonia including coronavirus pneumonia on a base of clinical symptomatology or an uncontrasted CT scan with suspicious images, were admitted to isolation departments at Cheikh Zaid International University Hospital in Rabat. Details of the clinical characteristics of these patients are summarized in Table 3.

### 3.4. Experimental Settings

The network was trained on Google Colab. All training and testing phases were performed in the same environment, using Keras deep learning framework and Python 3.5 as the programming language. The network training is performed with the hyperparameters illustrated in Table 4. Both datasets are divided into a training set and a testing set with a ratio of 0.7:0.3, respectively.

## 4. Experimental Results

In this section, experimental results on both public and private datasets are illustrated. The performance of the classification model was evaluated based on different metrics including, accuracy, confusion matrix, precision, recall and F1-score.

### 4.1. Results on X-ray Modality

The confusion matrix is used to determine the performance of the classification models for a given set of test data. It can only be determined if the true values for test data are known. According to Figure 5, the outcomes of the proposed model demonstrated the highest performance of the EfficientNet model, in which only one image of pneumonia is misclassified as the normal lung. Also, Table 5 shows that this model can efficiently classify the three classes with the highest ratio of precision for common pneumonia and coronavirus pneumonia is 100% and 99% for normal cases. This result assures that the classification is performed correctly for the three classes on the X-ray modality.

ResNet125V2 [33], DenseNet121 [34], and EfficientNetB7 [30] are all deep neural network architectures that have been used for various computer vision tasks such as image classification, object detection, and semantic segmentation. To validate the use of EfficientNetB7 in pneumonia classification, we perform different experiments in terms of accuracy and execution time. Each of these architectures has its own strengths and weaknesses, and the choice of which one to use depends on the specific task and available computational resources. As shown in Table 6, EfficientNetB7 is the best architecture in terms of accuracy, but it takes more time to train and produce a classification. Training a deep learning model typically requires a significant amount of computational resources and can take a long time to complete, depending on the size and complexity of the model and the amount of training data. Although training time is important, we can save the model once the results are very significant. Then the model can be used for inference. The inference time of EfficientNetB7 is about 2.51 s, and it refers to the classification of 540 X-ray images. On average, each image can be classified in 4 ms.

### 4.2. Results on CT Scan Modality

Furthermore, training and testing of the suggested network were done on the CT scan dataset. The confusion matrix is an important metric for the performance evaluations of a classification model. As depicted in Figure 6, only three normal images were confused with coronavirus pneumonia classes. From the values of Table 7, a precision of 99% was achieved for coronavirus pneumonia and 100% for both normal and common pneumonia. Also, this model attained 100% of recall for coronavirus pneumonia and common pneumonia classes. In this work, one of the main findings is that the proposed model can distinguish correctly between the lungs infected with coronavirus pneumonia and all the patients infected with common pneumonia.

Table 8 depicts comparison results by CT scan modality. We can assert that EfficienNetB7 is the best architecture for pneumonia classification. The inference time of EfficientNetB7 on CT scan images is about 41.50 s, and it refers to the classification of 2953 X-ray images. On average, each CT scan can be classified in 14 ms.

### 4.3. Results on Cheikh Zaid Data

All validation DICOM images taken at the Cheikh Zaid International University Hospital were analyzed and reviewed by a radiologist, who overlooked epidemiological history and clinical symptoms. He classified chest CT as normal, positive for coronavirus pneumonia, or positive for common pneumonia.

All Cheikh Zaid images were treated following the preprocessing pipeline. As shown in Figure 7, cropping of the original image was performed to remove the unwanted regions, then to improve the image quality, the histogram equalization technique was used. After that, the image was resized to be suitable for the input model. To perform classification, the trained model was loaded and the class of each Cheikh Zaid image was predicted. As a result, shown in Table 9, the proposed model achieved an accuracy of 95% on Moroccan CT scan images. Thus, it will help the radiologist to classify pneumonia diseases for further diagnosis.

### 4.4. Comparison with the State-of-the-Art

Table 10 compares the proposed model based on the EfficientNet architecture as well as the preprocessing pipeline with the state-of-the-art. Previous approaches have used the X-ray dataset or CT scan dataset to attain coronavirus pneumonia classification with data augmentation techniques and transfer learning. A high accuracy Was achieved on both chest X-ray and CT scan modalities. These results are very encouraging and can be used by radiologists to diagnose pneumonia diseases early and correctly.

## 5. Discussion

The proposed EfficientNet model showed higher classification performance on the three classes with the highest accuracy ratio on public datasets. The results of the present work only show that the proposed CADx system achieved high accuracy in public datasets for both X-ray and CT scans. For radiographic modality, Table 5 shows that the model efficiently and correctly classified the three classes with the highest accuracy ratio, which is 100% for common pneumonia and coronavirus pneumonia and 99% for normal cases. Regarding the CT scan modality and based on Table 6 values, an accuracy of 99% was obtained for coronavirus pneumonia and 100% for normal and common pneumonia. Similarly, this model achieved 100% recall for coronavirus pneumonia and common pneumonia classes. Therefore, the proposed model can correctly distinguish lungs infected by coronavirus pneumonia from those infected by common pneumonia. As a result, the proposed model achieved an accuracy of 95% on Moroccan CT images. This may occur in the presence of low-quality images that contain artifacts and noise similar to the opacity of patients’ lungs. Moreover, the characteristics of public datasets may be different from those of Moroccan clinical data. In such a case, an overfit may have occurred during external validation. In addition, the specificity of the proposed model can be summed up in the fact that it has achieved very satisfactory accuracy rates on two imaging modalities based on public data. This shows its robustness when compared with models cited in related works that have treated a single imaging modality. Therefore, its use will be of great help to radiologists in classifying pneumonia diseases for further diagnosis.

## 6. Conclusions

In this work, a Post-Pandemic Classification Algorithm (PPCA) based on the EfficientNet architecture is proposed in the context of pneumonia virus classification. The proposed approach consists of two major phases. In the first phase, preprocessing is performed on both X-ray and CT scan images. Whereas, in the second phase, the used architecture is trained on two public datasets as well as the Cheikh Zaid dataset. In sum, this work was able to implement an accurate computer-aided system for the analysis of both radiographic and CT medical images that led to more promising classification results and higher accuracy. Thus, this system will provide valuable support to radiologists and physicians for the early detection and diagnosis of respiratory viruses.

## Figures and Tables

**Figure 1 healthcare-11-00662-f001:**
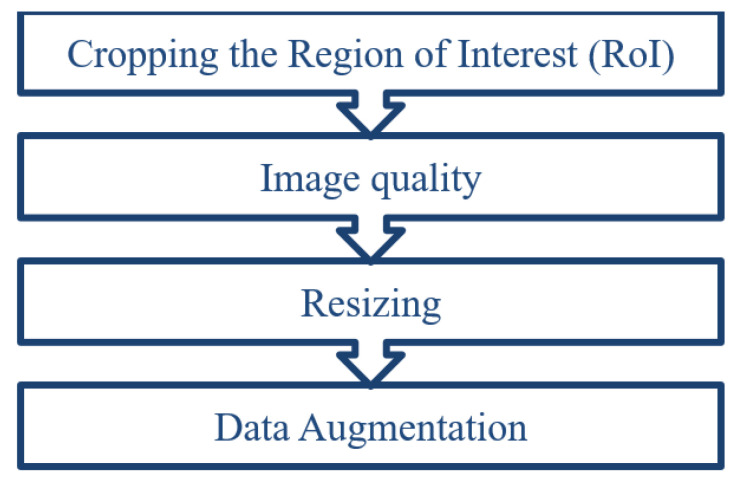
Preprocessing steps.

**Figure 2 healthcare-11-00662-f002:**
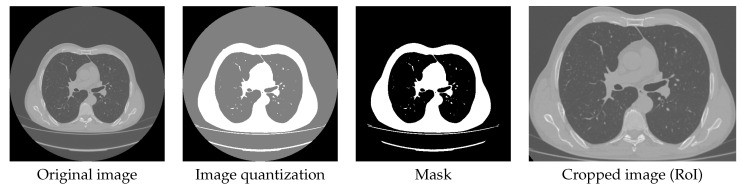
Preprocessing on Cheikh Zaid data.

**Figure 3 healthcare-11-00662-f003:**
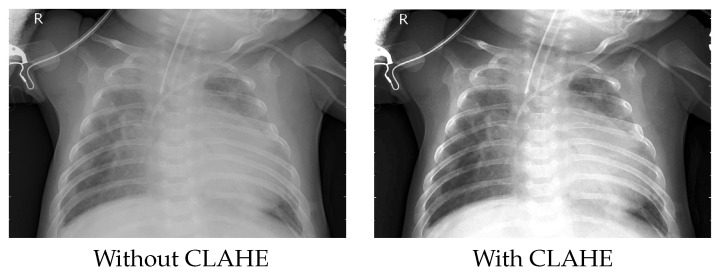
Contrast enhancement on the X-ray image.

**Figure 4 healthcare-11-00662-f004:**
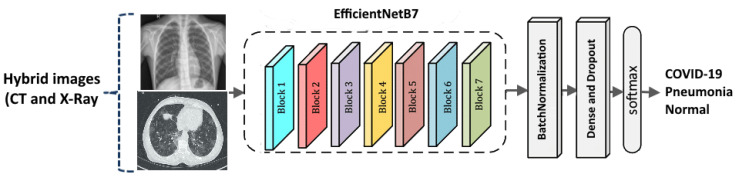
The proposed approach.

**Figure 5 healthcare-11-00662-f005:**
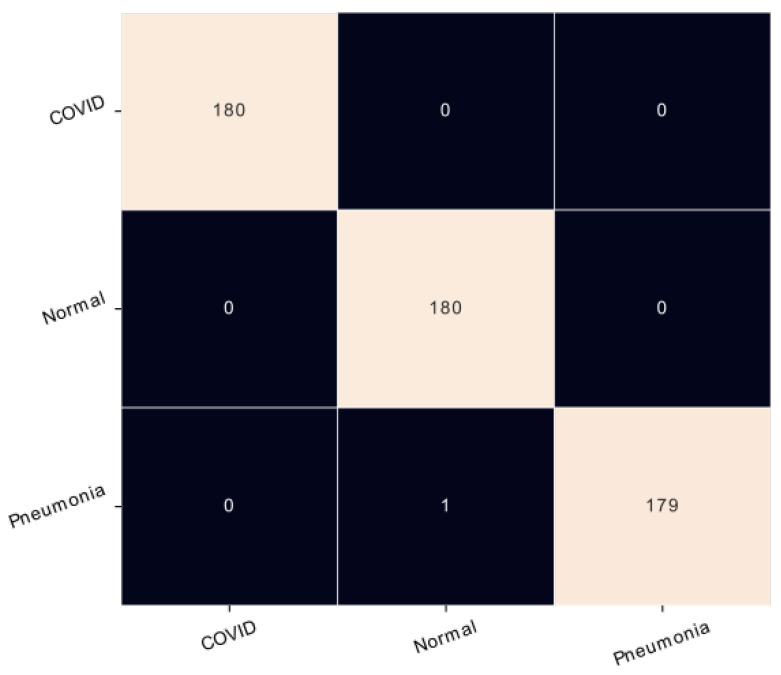
Confusion matrix of X-ray modality.

**Figure 6 healthcare-11-00662-f006:**
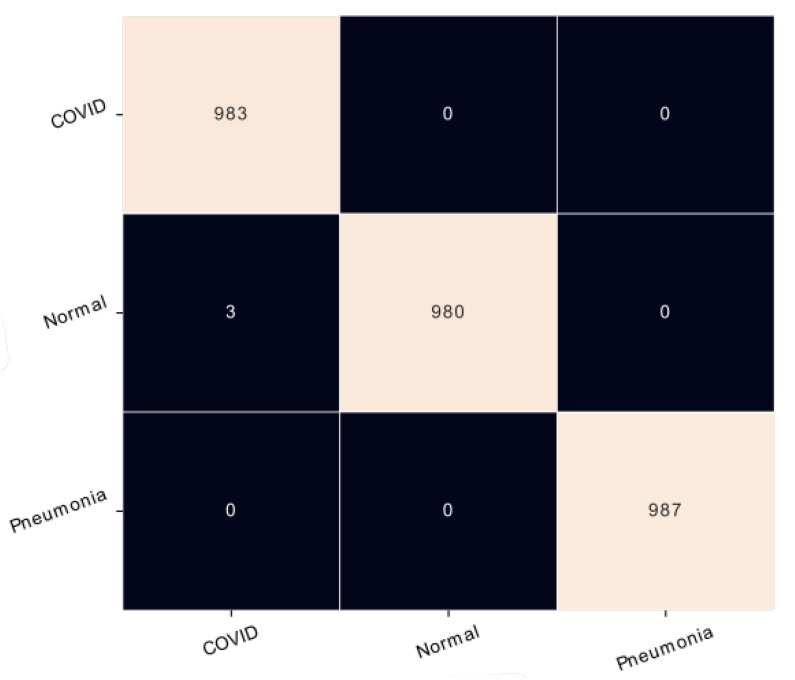
Confusion matrix of CT scan modality.

**Figure 7 healthcare-11-00662-f007:**
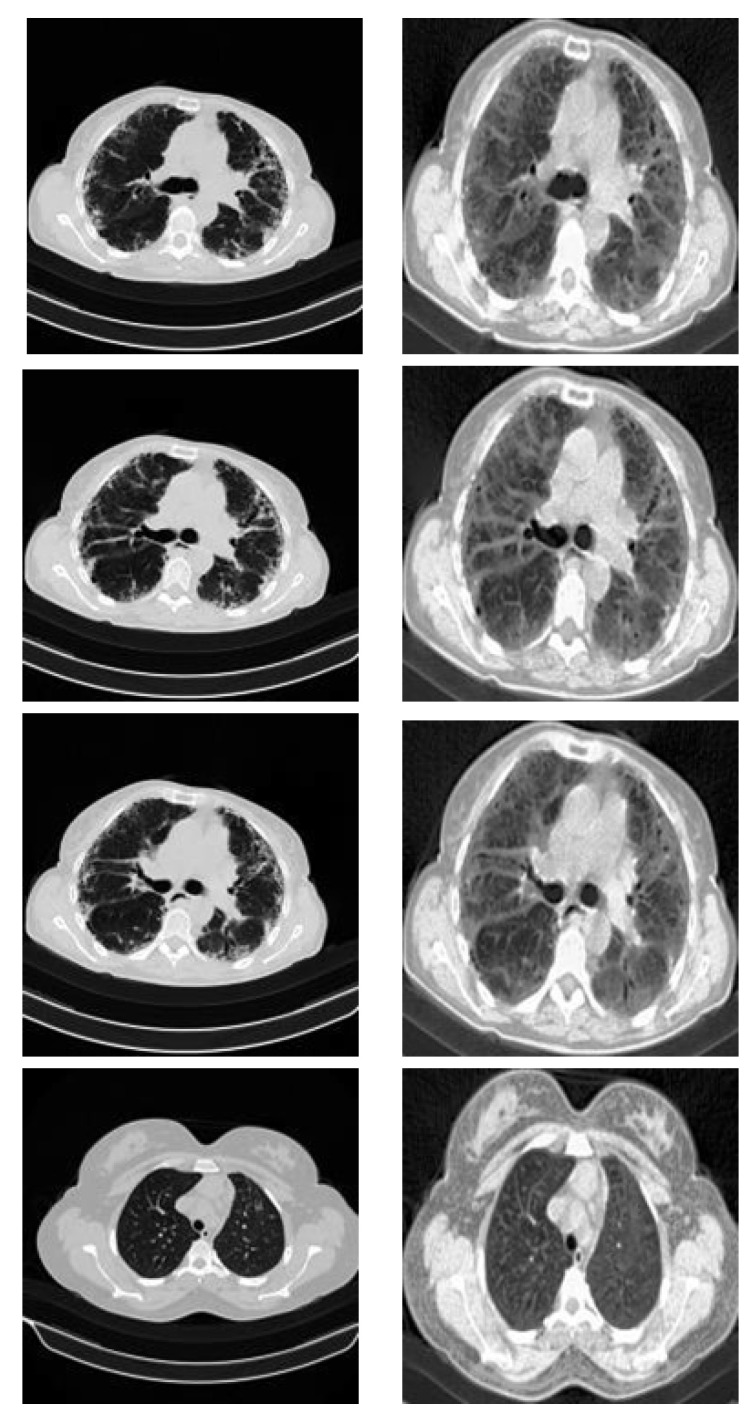
Preprocessing of Cheikh Zaid data. (**Left**) Before preprocessing. (**Right**) After preprocessing.

**Table 1 healthcare-11-00662-t001:** Datasets, architectures, methods and accuracy of X-ray-based approaches.

Article	Dataset	Architecture	Methods	Classes	Accuracy
Tripathi et al. [19]	Chest X-Ray 14	CNN+VGG+STN	DL	3	89.77%
Zhang et al. [20]	CXR dataset	Deep CNN	DL	2	95.2%
Sarki et al. [21]	CIDC	VGG16	TL	2	100%
				3	87.5%
		Proposed CNN	DL	2	97.67%
				3	93.75%
Tuncer et al. [22]	Undefined	F-transform, MKLBP and SVM	F-transform and MKLBP	3	97.01%
Nishio et al. [16]	Public datasets	VGG16	DL and TL	3	83.6%
		Proposed CNN			Less than 80%
Asif et al. [5]	COVID-chest Xray-dataset	Inception-v3	TL	3	More than 98%
Shelke et al. [23]	India Images	VGG16	DL	3	95.9%
		denseNet161		2	98.9%
		ResNet-18		3	76%
Ucar and Korkmaz [17]	Public dataset named COVIDx	COVIDiagnosis-Net, based on SqueezeNet	D L	3	98.3%
ElAraby et al. [24]	CXR	GSEN	DL + Crawler 2	2	95.60%
				3	92.76%

**Table 2 healthcare-11-00662-t002:** Datasets, architectures, methods, and accuracy of CT scan-based approaches (CoP and NCoP stand for COVID-19 Pneumonia and Non-COVID 19 Pneumonia respectively).

Article	Dataset	Architecture	Methods	Classes	Accuracy (%)
Saba et al. [25]	Covid dataset	iCNN & CNN	DL	2	99.30% & 99.53%
		VGG19 & IV3	TL	2	99.53% & 94.84%
		k-NN & RF	ML	2	96.84% & 74.58%
Shi et al. [26]	large-scale dataset CT scans	Proposed-CNN LR, SVM, NN	DL & ML	Groups with different ranges of infected lesion sizes	87.9%
Maftouni et al. [18]	Covid-CT dataset	DenseNet-121	DL	3	94.42%
		Residual Attention92		3	90.47%
		proposed with FC		3	98.32%
		proposed with FC+VM		3	97.93%
Nguyen et al. [27]	CC-CCII Dataset-China	CNN	DL	2 (Covid+ & Covid-)	87%/82.6% (UTSW)
	Covid-CTset-Iran				97%/98.8% (CC-CCCI)
	MosMedData-Russa				86%/87.3% Covid-CTset)
Pathan et al. [28]	Various sources	Proposed CNN: ResNet-50, AlexNet, VGG19, Densenet & Inception V3	TL	2(Covid+ & Covid-)	96%

**Table 3 healthcare-11-00662-t003:** Details of the clinical characteristics of patients.

	Non-Coronavirus Pneumonia (n = 47)	Coronavirus Pneumonia (n = 51)	Pneumonia (n = 29)
Age (year):			
<20	4 (8.51%)	7 (13.73%)	3 (10.34%)
20-39	16 (34.04%)	19 (37.25%)	11 (37.93%)
40-59	19 (40.43%)	17 (33.33%)	09 (31.04%)
≥60	08 (17.02%)	08 (15.69%)	06 (20.69%)
Sex:			
Male	26 (55.32%)	23 (45.10%)	18 (62.07%)
Female	21 (44.68%)	28 (54.90%)	11 (37.93%)
Presence of Fever:			
Fever	34 (72.34%)	46 (90.19%)	23 (79.31%)
No fever	13 (27.66%)	5 (9.81%)	6 (20.69%)
White blood cell Count:			
Normal	11 (23.41%)	3 (5.88%)	5 (17.24%)
Elevated	36 (76.59%)	48 (94.12%)	24 (82.76%)
Lymphocyte count:			
Normal	39 (82.97%)	9 (17.65%)	11 (37.93%)
Decreased	8 (17.03%)	42 (82.35%)	18 (62.07%)
Comorbidities:			
Cardiovascular Disease	3 (6.38%)	9 (17.64%)	6 (20.69%)
Hypertension	8 (17.02%)	13 (25.49%)	11 (37.93%)
COPD	6 (12.76%)	7 (13.72%)	4 (13.79%)
Diabetes	5 (10.64%)	12 (23.53%)	8 (27.59%)
Chronic liver Disease	1 (2.13%)	0 (0%)	0 (0%)
Chronic kidney Disease	0 (0%)	1 (1.96%)	0 (0%)
Malignant tumor	0 (0%)	2 (3.92%)	0 (0%)
HIV	0 (0%)	0 (0%)	0 (0%)
Severity:			
Mild	-	29 (57.7%)	-
Medium	-	16 (15.5%)	-
Severe	-	4 (16.3%)	-
Critical	-	2 (10%)	-

**Table 4 healthcare-11-00662-t004:** Hyperparameters maintained during training.

Parameter	Value
Input size	256×256
Batch size	8
Learning rate	$10^{-4}$
Optimizer	Adam
Epochs	100
Loss function	Categorical Crossentropy
Kernel initializer	Orthogonal

**Table 5 healthcare-11-00662-t005:** The classification report of our proposed approach on X-ray modality.

Classes	Precision	Recall	F1-Score
COVID-19	100%	100%	100%
Normal	99%	100%	100%
Pneumonia	100%	99%	100%

**Table 6 healthcare-11-00662-t006:** Classification comparison on X-ray modality using different architectures in terms of accuracy and execution time.

Architecture	Accuracy	Training Time (in s)	Inference Time (in s)
DenseNet121	97.73%	1229.04	2.44
ResNet152V2	95.18%	1647.44	5.48
EfficientNetB7	99.81%	1366.84	2.51

**Table 7 healthcare-11-00662-t007:** The classification report of our proposed approach on CT scan modality.

Classes	Precision	Recall	F1-Score
COVID-19	99%	100%	100%
Normal	100%	99%	100%
Pneumonia	100%	100%	100%

**Table 8 healthcare-11-00662-t008:** Classification comparison on CT scan modality using different architectures in terms of accuracy and execution time.

Architecture	Accuracy	Training Time (in s)	Inference Time (in s)
DenseNet121	92.63%	20,330.95	40.76
ResNet152V2	97.83%	15,897.32	30.87
EfficientNetB7	99.88%	21,960.07	41.50

**Table 9 healthcare-11-00662-t009:** The classification report of our proposed approach on Cheikh Zaid data.

Classes	Precision	Recall	F1-Score
Pneumonia	95%	93%	93%
COVID-19	94%	95%	94%
Normal	96%	97%	97%

**Table 10 healthcare-11-00662-t010:** Comparison with the state-of-the-art.

Approach	CT Scan Modality	X-ray Modality
Tripathi et al. [19]	–	89.77%
Sarki et al. [21]	–	93.75%
Tuncer et al. [22]	–	97.01%
Nishio et al. [16]	–	83.6%
Asif et al. [5]	–	98%
Shelke et al. [23]	–	95.9%
Ucar and Korkmaz [17]	–	98.3%
ElAraby et al. [24]	–	92.76%
Maftouni et al. [18]	98.32%	–
Our proposed approach	99.88%	99.81%

## Data Availability

No new data were created.

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
