# Peer review of "On the Implementation of a Post-Pandemic Deep Learning Algorithm Based on a Hybrid CT-Scan/X-ray Images Classification Applied to Pneumonia Categories"

_healthcare, 2023, doi:10.3390/healthcare11050662_

Round 1

Reviewer 1 Report

1.      Abstract need to be restructured

2.      Introduction lacks contribution and structure of what the other section will provide.

3.      The paper suggests some good models but lack serious comparison with existing model increase your literature and comparison with recent papers and compare models like

https://link.springer.com/article/10.1007/s11042-022-13551-2

https://link.springer.com/article/10.1007/s11042-021-11474-y,  https://link.springer.com/article/10.1007/s11042-022-12922-z,

these are some suggestions from my side.

4.      Abstract should reflect the background knowledge on the problem addressed need to be added.

5.      Abstract should reflect the wide range of applications and its possible solutions need to be added.

6.      Abstract should reflect the problem addressed need to be justified with more details.

7.      In Introduction section, the drawbacks of each conventional technique should be described clearly.

8.      Introduction section can be extended to add the issues with respect to existing work

9.      What is the motivation of the proposed work?

10.  Literature review techniques have to be strengthened by including the issues in the current system and how the author proposes to overcome the same

11.  Research gaps, objectives of the proposed work should be clearly justified.

Reviewer 2 Report

The research tried to propose a Computer Aided Diagnosis system based on deep learning model (EfficientNet) for detecting coronavirus pneumonia versus typical pneumonia and normal stage on X-ray as well as Computed tomography. The content is well structured and the presentation is good. Their effort on collecting the CT image from Cheikh Zaid hospital for validation data to evaluate their proposal is non-trivial and highly encouraged.

However, there are fault in their experiment design that potentially caused a biased result. The following pointed out are concerns the author should justify to get their manuscript be considered again.

1. The authors applied the data augmentation on the X-ray dataset thus generated 600 images per each class. The problem is that they split the train and set data (70%:30%) on augmented data which is 180 images per each class (according to the confusion matrix Figure 5). This design resulted in large amount of validation image, technically the augmented version of validation image, are in the train set which will lead to the biased evaluation. Definitely, the author must split the dataset first then apply the augmentation only to the train set for training then evaluate the trained model on the separated validation set without augmentation.

2. The CT dataset provider stated that the CT dataset (MOSMEDDATA) contains CT of COVID-19 (4 levels of severity: Mild, Moderate, Severe and Critical) and non-COVID-19 patients thus this dataset does not contain the CT image of Pneumonia. How can the authors train their 3 classes model with the dataset of 2 classes? Moreover, the distribution of CT dataset is 254, 684, 125, 45 and 2 for 4 levels of COVID-19 severity and non-COVID-19 respectively (total 1110 persons). How can the author manage to compile to a balanced 3 classes validation test which the distribution is 983, 983 and 987 for COVID, Normal and Pneumonia respectively? The author should clarify their experiment strategy/methodology/design with more detailed description.

3. As the CT system will take many slices (image) for one patient. What is the criteria the author has employed to select which slice of a patient to be the sample in their Cheikh Zaid Dataset?

Lastly, for transparent decision, the author should supply their experiment code so the reviewer can help to verify their experiment.

Reviewer 3 Report

The research work has no significant novelty component. Please find my comments:

1.       Novelty is limited. When data is downloaded and the model is pre-trained, the problem in not unique, and lots of work has been done already, then what is the novelty of the proposed work?

2.       Research gaps are limited.

3.       There are many typo mistakes in the paper, almost on every single page. Please check and rectify.

4.       It is mentioned, “It is composed of 603 X-ray images in JPEG format, 255 downloaded from Mendeley. 234 of these images were normal, 221 were positive for 256 coronavirus pneumonia and 141 of them were positive for pneumonia.” But the summation is not 603, please clarify.

5.       Table 8: Comparison is not proper. Authors can compare multiple number of parameters, i.e., number of classes, type of model, number of samples in the dataset etc.

6.       What are the steps taken for avoiding overfitting?

7.       Is there any effect of class imbalance?

Round 2

Reviewer 3 Report

Even after having severe concerns from all three reviewers, the authors have not incorporated the major changes in their manuscript. It is almost as it is in the previously submitted draft. The replies are not satisfactory, looking like the reviewer’s opinions are not necessary to implement.

1. There are many research works that have been done for more than three classes. Even a few of the mentioned papers are also pointing out more classes. For example, the first paper mentioned in Section 2, by Tripathi et al. [19] is for 15 categories of data. If such data are available with more classes, authors should consider this. Other authors should highlight the limitations of the work. So, novelty is not justified, unfortunately. The architecture and dataset-related contributions are very limited.

2. Authors have also mentioned their work is very rare where CT scans and X-ray images were used. But many research works are already published considering both modalities. Some of them are as follows:

a)       A deep learning and grad-CAM based color visualization approach for fast detection of COVID-19 cases using chest X-ray and CT-Scan images (H Panwar, PK Gupta, MK Siddiqui… - Chaos, Solitons & …, 2020 – Elsevier)

b)      Diagnostic accuracy of X-ray versus CT in COVID-19: a propensity-matched database study (A Borakati, A Perera, J Johnson, T Sood - BMJ open, 2020 - bmjopen.bmj.com)

c)       Comparing CT scan and chest X-ray imaging for COVID-19 diagnosis (E Benmalek, J Elmhamdi, A Jilbab - Biomedical Engineering Advances, 2021 – Elsevier)

d)      X-ray and CT-scan-based automated detection and classification of covid-19 using convolutional neural networks (CNN) (S Thakur, A Kumar - Biomedical Signal Processing and Control, 2021 – Elsevier)

e)      An overview of deep learning techniques on chest X-ray and CT scan identification of COVID-19 (WC Serena Low, JH Chuah, CATH Tee… - … Methods in Medicine, 2021 - hindawi.com)

f)        Computer aid screening of COVID-19 using X-ray and CT scan images: An inner comparison (PK Sethy, SK Behera, K Anitha… - Journal of X-ray …, 2021 - content.iospress.com)

g)       Automatic diagnosis of coronavirus (COVID-19) using shape and texture characteristics extracted from X-Ray and CT-Scan images (M Imani - Biomedical Signal Processing and Control, 2021 – Elsevier)

3. Authors should compare the performance of the EffecientNet-B7 with other deep learning models so that superiority can be justified.

4. Many of the responses are looking like personal replies, authors should understand that their excuses in the research works are not worthy enough in the current scenario where lots of work have been done and lots of data is available.
